# Design of a High-Order Kalman Filter for State and Measurement of A Class of Nonlinear Systems Based on Kronecker Product Augmented Dimension

**DOI:** 10.3390/s23062894

**Published:** 2023-03-07

**Authors:** Deyan Peng, Chenglin Wen, Meilei Lv

**Affiliations:** 1School of Automation, Hangzhou Dianzi University, Hangzhou 310018, China; 2College of Electrical and Information Engineering, Quzhou University, Quzhou 324000, China; 3School of Automation, Guangdong University of Petrochemical Technology, Maoming 525000, China

**Keywords:** full-dimensional variables, Taylor expansion, Kronecker product, projection operator

## Abstract

A high-order Kalman filter for full-dimensional variables is proposed for a class of dynamic systems whose state model and measurement model are both nonlinear. The filter requires Taylor expansion of the system equations, and then performs Kronecker product operation on the linear part in the Taylor expansion. Finally, a linear dynamic model is achieved based on the full-dimensional vector formed by the state variables and the high-order dimension expansion variables. After designing the filter, the Kalman filter for the original state variables estimation was selected through the projection operator. The excellent performance of the novel filter is analyzed from the aspects of the information utilization of the state estimation value and the size of the state estimation error covariance matrix. The numerical verification is carried out by computer simulation.

## 1. Introduction

In the fields of target tracking, unmanned driving, fault diagnosis, etc., an important indicator to judge the performance of the system is to determine whether the state estimation value of the system is accurate [1,2,3,4,5,6]. The filtering algorithm is commonly used in state estimation. In order to obtain an accurate state estimation, various filters have been studied by predecessors. Among them, the Kalman filter (KF) is the most representative. KF is an optimal filter based on the minimum variance criterion under the linear Gaussian system [7,8]. However, most of the real systems are nonlinear, and the application of KF is limited. To obtain greater practical value in engineering applications, the researchers retreated to the next best thing and proposed the extended Kalman filter (EKF). EKF has been proven to be an effective filter in weakly nonlinear systems and has a wide range of application scenarios [9,10]. However, EKF performs Taylor expansion on nonlinear functions, only the first-order linear polynomial items are retained, and all other high-order polynomial items are discarded. Due to the introduction of a large number of model rounding errors, EKF filtering performance is often reduced or even diverged. To solve this problem, an unscented Kalman filter (UKF) and a cubature Kalman filter (CKF) have been proposed. The core idea of UKF is unscented transformation (UT), which uses the set of sampling points to approximate the nonlinear system [11,12]. CKF optimizes the sampling method and weight distribution of the sampling points in UKF through a spherical integral and a radial integral [13]. Both UKF and CKF can achieve the second-order polynomial approximation ability for nonlinear systems [14]. However, as the degree of nonlinearity of the model increases, not only will the performance of the filter decrease, but it also often causes the filter to diverge. At the same time, due to improper selection of weights in UKF and CKF, the calculated prediction error appears as a non-negative definite phenomenon, resulting in filter overflow.

KF, EKF, UKF, and CKF are only applicable to Gaussian noise, therefore approximate analytical solutions can be obtained regardless of whether the model is linear or not. However, a particle filter (PF) has no limitation of the model or Gaussian noise. It directly calculates the mean value of posterior probability through weighted random samples. However, PF still faces the problem of a lack of particle diversity and high computational complexity [15]. In order to improve the applicability of the KF method in the nonlinear non-Gaussian system, reference [16] proposed the characteristic function filter (CFF) by replacing the probability density function with the characteristic function. However, the proposal and development of CFF are currently only designed for systems where the state equation is linear and the measurement equation is generally nonlinear [17]. The scope of application is still limited.

After CKF was proposed, the research on EKF did not make substantial progress until the emergence of the polynomial extended Kalman filter (PEKF). PEKF approximates the nonlinear system by the polynomial, which can be based on the nonlinear function form of the system and the estimated performance index. It can achieve higher accuracy than EKF, but the nonlinear form of PEKF expansion is complex and difficult to understand. Not only does each step require complex Kronecker product operation on the original system, but complex high-order Taylor expansion and more complex multivariate binomial expansion based on Taylor expansion [18,19,20,21,22,23,24] are also required. The Kronecker product operation can transform the two matrices into a larger matrix, which contains all possible products of the two matrices. It is often used to solve problems in linear algebra and its applications. For example, reference [25] introduced the Kronecker product in speech processing and expressed the linear prediction coefficients as two sub-filter coefficient vectors, which improves the computational efficiency and speech quality. In addition, the nonlinear filter of the linear parameter was proposed by the Kronecker integral solution in reference [26], and it can provide better noise reduction capabilities in nonlinear active noise control scenarios. Reference [27] proposed an iterative multi-channel Wiener filter based on the Kronecker integral solution of the impulse response in the multi-input and single-output system problem, which can have good accuracy when the amount of statistically estimated data is small.

In order to solve the computational complexity of PEKF, the high-order extended Kalman filter (HEKF) was proposed for a class of strongly nonlinear systems [28]. HEKF only performs high-order Taylor expansion similar to PEKF on the original state model and defines all high-order polynomial terms as hidden state variables of the corresponding order. A simplified version of PEKF is established by establishing a random walk model of the hidden variables. Compared with PEKF, the computational complexity of HEKF is greatly reduced. However, because it does not use any prior information from the original state model, the established model of extended dimension is relatively rough, and its performance also greatly suffered a large loss.

Therefore, in order to overcome the above-mentioned shortcomings of the high precision but high complexity of PEKF, and the low complexity but low estimation accuracy of HEKF, this paper proposes a compromise-friendly filter design method, which is for both PEKF and HEKF. In the high-order Taylor expansion polynomial of the original model, only the linear part of the state and measurement equations is used to model by the Kronecker product. It can not only reduce the complexity of the problem, but also effectively improve the accuracy of the estimation. The main work of this paper is: (1) firstly, the first-order Taylor expansion of the system state equation and measurement equation is carried out to obtain the approximate linearized equation. (2) Then, the items of the approximate linearized state equation are respectively operated by Kronecker product. The dimension of the model can be expanded, and the hidden variables are introduced. Then, the items of the approximate linearized measurement equation (except the noise term) are also operated through the Kronecker product. Together with the expanded state equation, a complete standard Kalman filter form is established. (3) After obtaining the expanded dimension of the system equation, the standard Kalman filter method is used to filter. Finally, after projecting through the projection matrix, only the estimated value of the original system state is retained. Therefore, the complexity of the algorithm is reduced, more information from the model is included, and the estimated accuracy is improved.

The remainder of this paper is organized as follows: the second section briefly describes the nonlinear systems; the third section gives the design idea and design method steps of Kalman filter based on the Kronecker product; the fourth section verifies the effectiveness of the proposed method through simulation examples; and the fifth section gives a summary of this paper.

## 2. Description of Nonlinear System

The state equation and measurement equation of the nonlinear system is given:(1)x(k+1)=f(x(k))+w(k)
(2)y(k+1)=h(x(k+1))+v(k+1)
where $x(k)\in R^{n}$ is the system state; $y(k+1)\in R^{m}$ is the measured output; and f(x(k)) and h(x(k+1)) are time continuously differentiable nonlinear maps, indicating the state transition function and measurement function of the system, respectively. The state noise w(k) and the output noise v(k+1) are assumed to be uncorrelated white noise sequences with zero mean. The noises satisfy the following statistical characteristics:E{w(k)}=0, $E\{w(k)w{\left(j\right)}^{T}\}=Q(k)\delta_{kj}$; E{v(k+1)}=0, $E\{v(k+1)v^{T}(j+1)\}=R(k+1)\delta_{kj}$. The process noise variance Q(k) is a positive semidefinite matrix, and the measurement noise variance R(k+1) is a positive definite matrix. $\delta_{kj}$ is a Dirac function and if k=j, $\delta_{kj}=1$, otherwise $\delta_{kj}=0$. The initial state x(0) is a random variable independent of the noise sequences. It is assumed that $E\{x(0)\}=x_{0}$, $E\{[x(0)-x_{0}]{\left[x(0)-x_{0}\right]}^{T}\}=P_{0}$, $E\{x(0)w^{T}(k)\}=0$, $E\{x(0)v^{T}(k+1)\}=0$, $P_{0}$ is a positive definite matrix.

## 3. Nonlinear System Filter Design Based on Kronecker Product

In this paper, the designed filter needs to use first-order Taylor expansion and a Kronecker product. Using the Kronecker product on the Kalman filter can achieve the expansion of the state dimension. Through the extended dimension, it can avoid losing too much information due to the existence of truncation error in the approximate linearized process, so that more useful information can be obtained by using high-order nonlinear characteristics in state estimation. Ultimately, the estimation accuracy is improved. First, the first-order Taylor expansion of the system equation is performed to obtain the Jacobian matrix and the fixed deviation term. After obtaining the linearized form, the Kronecker product operation is performed on the terms of the approximate state equation and the terms of the approximate measurement equation (except for the noise term). The operation of extended dimension is performed on the entire system equation. Finally, after projecting through the projection matrix, only the state estimation value of the original system is retained.
**Theorem** **1.***Let M and N be matrices of dimensions r×s and p×q, respectively. Then the Kronecker product M⊗N is defined as the (r·p)×(q·s) matrix [29,30]*(3)$$M⊗N=\left[\begin{array}{c} m_{11}N\;\cdots \cdots \;m_{1s}N\\ \cdots \cdots \cdots \cdots \cdots \cdots \cdots \\ m_{r1}N\;\cdots \cdots \;m_{rs}N \end{array}\right]$$*where* $m_{ij}$ *are the entries of*
 M
*. Of course, this kind of product is not commutative.*

Moreover, the Kronecker power of M is defined as:(4)$$M^{\left[0\right]}=1\in R$$
(5)$$M^{\left[r\right]}=M⊗M^{\left[r-1\right]},r\geq 1$$


(1)First, the nonlinear function f(x(k)) in Equation (1) is first-order Taylor expanded around the nominal state $\hat{x}(k|k)$:(6)$$x(k+1)\approx f(\hat{x}(k|k))+{\frac{\partial f(x(k))}{\partial x(k)}|}_{x(k)=\hat{x}(k|k)}[x(k)-\hat{x}(k|k)]+w(k)$$


The approximate linearization equation of the state equation can be obtained:(7)$$x(k+1)=A(k+1,k)x(k)+\bar{f}(\hat{x}(k|k))+w(k)$$
where
(8)$$A(k+1,k)={\frac{\partial f(x(k))}{\partial x(k)}|}_{x(k)=\hat{x}(k|k)}={\left[\begin{array}{ccc} \frac{\partial f_{1}(x(k))}{\partial x_{1}(k)} & \cdots & \frac{\partial f_{1}(x(k))}{\partial x_{n}(k)}\\ \vdots & \ddots & \vdots \\ \frac{\partial f_{n}(x(k))}{\partial x_{1}(k)} & \cdots & \frac{\partial f_{n}(x(k))}{\partial x_{n}(k)} \end{array}\right]|}_{x(k)=\hat{x}(k|k)}$$
(9)$$\bar{f}(\hat{x}(k|k))=f(\hat{x}(k|k))-{\frac{\partial f(x(k))}{\partial x(k)}|}_{x(k)=\hat{x}(k|k)}\cdot \hat{x}(k|k)$$

The nonlinear function h(x(k+1)) in Equation (2) is first-order Taylor expanded around the state x(k+1):(10)$$y(k+1)\approx h(\hat{x}(k+1|k))+{\frac{\partial h(x(k+1))}{\partial x(k+1)}|}_{x(k)=\hat{x}(k+1|k)}[x(k+1)-\hat{x}(k+1|k)]+v(k+1)$$

The approximate linearized equation of the measured equation can be obtained:(11)$$y(k+1)=H(k+1)x(k+1)+\bar{h}(\hat{x}(k+1|k))+v(k+1)$$
where
(12)$$H(k+1)={\frac{\partial h(x(k+1))}{\partial x(k+1)}|}_{x(k+1)=\hat{x}(k+1|k)}={\left[\begin{array}{ccc} \frac{\partial h_{1}(x(k+1))}{\partial x_{1}(k+1)} & \cdots & \frac{\partial h_{1}(x(k+1))}{\partial x_{m}(k+1)}\\ \vdots & \ddots & \vdots \\ \frac{\partial h_{m}(x(k+1))}{\partial x_{1}(k+1)} & \cdots & \frac{\partial h_{m}(x(k+1))}{\partial x_{m}(k+1)} \end{array}\right]|}_{x(k+1)=\hat{x}(k+1|k)}$$
(13)$$\bar{h}(\hat{x}(k+1|k))=h(\hat{x}(k+1|k))-{\frac{\partial h(x(k+1))}{\partial x(k+1)}|}_{x(k+1)=\hat{x}(k+1|k)}\cdot \hat{x}(k+1|k)$$
(2)The linear term, fixed deviation term, and noise term of Equation (7) are respectively subjected to Kronecker product operation to obtain the r-order state extended equation:
(14)$$x^{\left[r\right]}(k+1)=A^{\left[r\right]}(k+1,k)x^{\left[r\right]}(k)+{\bar{f}}^{\left[r\right]}(\hat{x}(k|k))+w^{\left[r\right]}(k)$$
where
(15)$$A^{\left[r\right]}(k+1,k)=\underset{r}{\underset{︸}{A(k+1,k)⊗A(k+1,k)⊗\cdots ⊗A(k+1,k)}}$$
(16)$${\bar{f}}^{\left[r\right]}(\hat{x}(k|k))=\underset{r}{\underset{︸}{\bar{f}(\hat{x}(k|k))⊗\bar{f}(\hat{x}(k|k))⊗\cdots ⊗\bar{f}(\hat{x}(k|k))}}$$
(17)$$w^{\left[r\right]}(k)=\underset{r}{\underset{︸}{w(k)⊗w(k)⊗\cdots ⊗w(k)}}$$
(18)$$\begin{array}{ll} x^{\left[r\right]}(k+1) & =\underset{r}{\underset{︸}{x(k+1)⊗x(k+1)⊗\cdots ⊗x(k+1)}}\\ & =\underset{r}{\underset{︸}{\left[\begin{array}{c} x_{1}(k+1)\\ x_{2}(k+1)\\ \vdots \\ x_{n}(k+1) \end{array}\right]⊗\left[\begin{array}{c} x_{1}(k+1)\\ x_{2}(k+1)\\ \vdots \\ x_{n}(k+1) \end{array}\right]⊗\cdots ⊗\left[\begin{array}{c} x_{1}(k+1)\\ x_{2}(k+1)\\ \vdots \\ x_{n}(k+1) \end{array}\right]}}\\ & =\left[\begin{array}{c} \underset{r}{\underset{︸}{x_{1}(k+1)x_{1}(k+1)\cdots x_{1}(k+1)x_{1}(k+1)}}\\ \underset{r}{\underset{︸}{x_{1}(k+1)x_{1}(k+1)\cdots x_{1}(k+1)x_{2}(k+1)}}\\ \vdots \\ \underset{r}{\underset{︸}{x_{n}(k+1)x_{n}(k+1)\cdots x_{n}(k+1)x_{n}(k+1)}} \end{array}\right] \end{array}$$

Combining Equation (7) and Equation (14), the general form of the state equation of extended dimension can be obtained:(19)$$\begin{array}{l} \left[\begin{array}{c} x^{\left[1\right]}(k+1)\\ x^{\left[2\right]}(k+1)\\ \vdots \\ x^{\left[r\right]}(k+1) \end{array}\right]=\left[\begin{array}{cccc} A^{\left[1\right]}(k+1,k) & 0 & 0 & 0\\ 0 & A^{\left[2\right]}(k+1,k) & 0 & 0\\ 0 & 0 & \ddots & 0\\ 0 & 0 & 0 & A^{\left[r\right]}(k+1,k) \end{array}\right]\left[\begin{array}{c} x^{\left[1\right]}(k)\\ x^{\left[2\right]}(k)\\ \vdots \\ x^{\left[r\right]}(k) \end{array}\right]\\ \;+\left[\begin{array}{c} {\bar{f}}^{\left[1\right]}(\hat{x}(k|k))\\ {\bar{f}}^{\left[2\right]}(\hat{x}(k|k))\\ \vdots \\ {\bar{f}}^{\left[r\right]}(\hat{x}(k|k)) \end{array}\right]+\left[\begin{array}{c} w^{\left[1\right]}(k)\\ w^{\left[2\right]}(k)\\ \vdots \\ w^{\left[r\right]}(k) \end{array}\right] \end{array}$$

Set,
(20)$$\begin{array}{c} X(k+1)=\left[\begin{array}{c} x^{\left[1\right]}(k+1)\\ x^{\left[2\right]}(k+1)\\ \vdots \\ x^{\left[r\right]}(k+1) \end{array}\right],\bar{A}(k+1,k)=\left[\begin{array}{cccc} A^{\left[1\right]}(k+1,k) & 0 & 0 & 0\\ 0 & A^{\left[2\right]}(k+1,k) & 0 & 0\\ 0 & 0 & \ddots & 0\\ 0 & 0 & 0 & A^{\left[r\right]}(k+1,k) \end{array}\right]\\ \Delta f(\hat{x}(k|k))=\left[\begin{array}{c} {\bar{f}}^{\left[1\right]}(\hat{x}(k|k))\\ {\bar{f}}^{\left[2\right]}(\hat{x}(k|k))\\ \vdots \\ {\bar{f}}^{\left[r\right]}(\hat{x}(k|k)) \end{array}\right],W(k)=\left[\begin{array}{c} w^{\left[1\right]}(k)\\ w^{\left[2\right]}(k)\\ \vdots \\ w^{\left[r\right]}(k) \end{array}\right] \end{array}$$

Then the extended dimension equation of the state can be written as:(21)$$X(k+1)=\bar{A}(k+1,k)X(k)+\Delta f(\hat{x}(k|k))+W(k)$$
(3)The Kronecker product operation is performed on the linear term and the fixed deviation term of Equation (11), and the noise term is independently modeled to obtain the corresponding r-order measured equation of extended dimension:
(22)$$y^{\left[r\right]}(k+1)=H^{\left[r\right]}(k+1)x^{\left[r\right]}(k+1)+{\bar{h}}^{\left[r\right]}(\hat{x}(k+1|k))+v^{\left(r\right)}(k+1)$$
where
(23)$$\begin{array}{ll} y^{\left[r\right]}(k+1) & =\underset{r}{\underset{︸}{y(k+1)⊗y(k+1)⊗\cdots ⊗y(k+1)}}\\ & =\underset{r}{\underset{︸}{\left[\begin{array}{c} y_{1}(k+1)\\ y_{2}(k+1)\\ \vdots \\ y_{m}(k+1) \end{array}\right]⊗\left[\begin{array}{c} y_{1}(k+1)\\ y_{2}(k+1)\\ \vdots \\ y_{m}(k+1) \end{array}\right]⊗\cdots ⊗\left[\begin{array}{c} y_{1}(k+1)\\ y_{2}(k+1)\\ \vdots \\ y_{m}(k+1) \end{array}\right]}}\\ & =\left[\begin{array}{c} \underset{r}{\underset{︸}{y_{1}(k+1)y_{1}(k+1)\cdots y_{1}(k+1)y_{1}(k+1)}}\\ \underset{r}{\underset{︸}{y_{1}(k+1)y_{1}(k+1)\cdots y_{1}(k+1)y_{2}(k+1)}}\\ \vdots \\ \underset{r}{\underset{︸}{y_{m}(k+1)y_{m}(k+1)\cdots y_{m}(k+1)y_{m}(k+1)}} \end{array}\right] \end{array}$$
(24)$$H^{\left[r\right]}(k+1)=\underset{r}{\underset{︸}{H(k+1)⊗H(k+1)⊗\cdots ⊗H(k+1)}}$$
(25)$${\bar{h}}^{\left[r\right]}(\hat{x}(k+1|k))=\underset{r}{\underset{︸}{\bar{h}(\hat{x}(k+1|k))⊗\bar{h}(\hat{x}(k+1|k))⊗\cdots ⊗\bar{h}(\hat{x}(k+1|k))}}$$

Combining Equation (11) with Equation (22), the general form of the measured equation of the extended dimension is obtained:(26)$$\begin{array}{l} \left[\begin{array}{c} y^{\left[1\right]}(k+1)\\ y^{\left[2\right]}(k+1)\\ \vdots \\ y^{\left[r\right]}(k+1) \end{array}\right]=\left[\begin{array}{cccc} H^{\left[1\right]}(k+1) & 0 & 0 & 0\\ 0 & H^{\left[2\right]}(k+1) & 0 & 0\\ 0 & 0 & \ddots & 0\\ 0 & 0 & 0 & H^{\left[r\right]}(k+1) \end{array}\right]\left[\begin{array}{c} x^{\left[1\right]}(k+1)\\ x^{\left[2\right]}(k+1)\\ \vdots \\ x^{\left[r\right]}(k+1) \end{array}\right]\\ \;+\left[\begin{array}{c} {\bar{h}}^{\left[1\right]}(\hat{x}(k+1|k))\\ {\bar{h}}^{\left[2\right]}(\hat{x}(k+1|k))\\ \vdots \\ {\bar{h}}^{\left[r\right]}(\hat{x}(k+1|k)) \end{array}\right]+\left[\begin{array}{c} v^{\left(1\right)}(k+1)\\ v^{\left(2\right)}(k+1)\\ \vdots \\ v^{\left(r\right)}(k+1) \end{array}\right] \end{array}$$

Set,
(27)$$\begin{array}{c} Z(k+1)=\left[\begin{array}{c} y^{\left[1\right]}(k+1)\\ y^{\left[2\right]}(k+1)\\ \vdots \\ y^{\left[r\right]}(k+1) \end{array}\right],\bar{H}(k+1)=\left[\begin{array}{cccc} H^{\left[1\right]}(k+1) & 0 & 0 & 0\\ 0 & H^{\left[2\right]}(k+1) & 0 & 0\\ 0 & 0 & \ddots & 0\\ 0 & 0 & 0 & H^{\left[r\right]}(k+1) \end{array}\right]\\ \Delta h(\hat{x}(k+1|k))=\left[\begin{array}{c} {\bar{h}}^{\left[1\right]}(\hat{x}(k+1|k))\\ {\bar{h}}^{\left[2\right]}(\hat{x}(k+1|k))\\ \vdots \\ {\bar{h}}^{\left[r\right]}(\hat{x}(k+1|k)) \end{array}\right],V(k+1)=\left[\begin{array}{c} v^{\left(1\right)}(k+1)\\ v^{\left(2\right)}(k+1)\\ \vdots \\ v^{\left(r\right)}(k+1) \end{array}\right] \end{array}$$

Then the measured equation of the extended dimension can be written as:(28)$$Z(k+1)=\bar{H}(k+1)X(k+1)+\Delta h(\hat{x}(k+1|k))+V(k+1)$$

(4)Combining Equations (21) and (28), the systematic equation after the extending dimension can be obtained:


(29)
$$\begin{array}{l} X(k+1)=\bar{A}(k+1,k)X(k)+\Delta f(\hat{x}(k|k))+W(k)\\ Z(k+1)=\bar{H}(k+1)X(k+1)+\Delta h(\hat{x}(k+1|k))+V(k+1) \end{array}$$


The statistical characteristics of the system after the extending dimension:(30)$$\begin{array}{l} E\{W(k)W^{T}(k)\}=Q_{W}(k)\\ E\{V(k+1)V^{T}(k+1)\}=R_{V}(k+1)\\ E\{W(k)V^{T}(k+1)\}=0 \end{array}$$

(5)The steps of the designed filter algorithm are given below.

**Step 1:** Set the initial values for the new system.

The initial value x(k) of the original state satisfies:(31)$$\begin{array}{l} E\{x(0)\}={\hat{x}}_{0}={\left[{\hat{x}}_{10},{\hat{x}}_{20},\cdots ,{\hat{x}}_{n0}\right]}^{T}\\ E\{[x(0)-x_{0}]{\left[x(0)-x_{0}\right]}^{T}\}=P_{0}=diag\{[p_{1,0},p_{2,0},\cdots ,p_{n,0}]\} \end{array}$$

The new system satisfies the following characteristics:(32)$$\begin{array}{l} X(0)={\left[{\left(x^{\left(1\right)}(0)\right)}^{T},{\left(x^{\left(2\right)}(0)\right)}^{T},\;\cdots \;,{\left(x^{\left(s\right)}(0)\right)}^{T}\right]}^{T}\\ E\{X(0)\}={\hat{X}}_{0}={\left[{\left({\hat{x}}_{0}^{\left(1\right)}\right)}^{T},{\left({\hat{x}}_{0}^{\left(2\right)}\right)}^{T},\;\cdots \;,{\left({\hat{x}}_{0}^{\left(s\right)}\right)}^{T}\right]}^{T}\\ E\{[X(0)-{\hat{X}}_{0}]\}={\bar{P}}_{0}=diag\{[P_{0}^{\left(1\right)},P_{0}^{\left(2\right)},\;\cdots \;,P_{0}^{\left(s\right)}]\}\\ \mathrm{where},s=n+n^{2}+\cdots +n^{r} \end{array}$$

**Step 2:** Recursive filtering

Assuming that $\hat{X}(k|k)$ and $\bar{P}(k|k)$ of the new system is known, the new Kalman filter is designed as follows:(33)$$\begin{array}{ll} \hat{X}(k+1|k+1) & =E\{X(k+1)|{\hat{X}}_{0},y(1),y(2),\cdots ,y(k),y(k+1)\}\\ & =E\{X(k+1)|\hat{X}(k|k),y(k+1)\} \end{array}$$

The corresponding estimated error covariance matrix:(34)$$\bar{P}(k+1|k+1)=E\{[X(k+1)-\hat{X}(k+1|k+1)]{\left[X(k+1)-\hat{X}(k+1|k+1)\right]}^{T}\}$$

**Step 3:** Time update

According to $\hat{X}(k|k)$, $\bar{A}(k+1,k)$ and $\Delta f(\hat{x}(k|k))$, the predicted value is obtained:(35)$$\hat{X}(k+1|k)=\bar{A}(k+1,k)\hat{X}(k|k)+\Delta f(\hat{x}(k|k))$$

According to $\bar{A}(k+1,k)$, $\bar{P}(k|k)$ and $Q_{W}(k)$, the predicted error covariance matrix is obtained:(36)$$\bar{P}(k+1|k)=\bar{A}(k+1,k)\bar{P}(k|k){\bar{A}}^{T}(k+1,k)+Q_{W}(k)$$

**Step 4:** Measurement update

According to $\bar{P}(k+1|k)$ and related measurement information, the Kalman gain can be obtained:(37)$$\bar{K}(k+1)=\bar{P}(k+1|k){\bar{H}}^{T}(k+1){\left[\bar{H}(k+1)\bar{P}(k+1|k){\bar{H}}^{T}(k+1)+R_{V}(k+1)\right]}^{-1}$$

According to $\hat{X}(k+1|k)$, $\bar{K}(k+1)$ and the actual measurement and estimated measured information, the estimated value of the new Kalman filter can be obtained:(38)$$\hat{X}(k+1|k+1)=\hat{X}(k+1|k)+\bar{K}(k+1)[Z(k+1)-\bar{H}(k+1)\hat{X}(k+1|k)]$$

The corresponding estimated error covariance matrix:(39)$$\bar{P}(k+1|k+1)=[I-\bar{K}(k+1)\bar{H}(K+1)]\bar{P}(k+1|k)$$

**Step 5:** Preserving the state estimate of the original system by the projection matrix.

Assuming that the projection matrix $p_{n}=[I_{n\times n},0_{n\times n^{2}},\cdots ,0_{n\times n^{r}}]$, where $I_{n\times n}$ and $0_{n\times n^{r}}$ are, respectively, the identity matrix of the corresponding dimension of the original state and the zero matrix of the corresponding dimension of the expanded state, the estimated value of the original system state is
(40)$$\begin{array}{l} \hat{x}(k+1|k+1)=p_{n}\cdot \hat{X}(k+1|k+1)=[I_{n\times n},0_{n\times n^{2}},\cdots ,0_{n\times n^{r}}]\hat{X}(k+1|k+1)\\ =[I_{n\times n},0_{n\times n^{2}},\cdots ,0_{n\times n^{r}}]\hat{X}(k+1|k)+[I_{n\times n},0_{n\times n^{2}},\cdots ,0_{n\times n^{r}}]\bar{K}(k+1)\tilde{Z}(k+1|k)\\ =\hat{x}(k+1|k)+K(k+1)\bar{H}(k+1)\tilde{X}(k+1|k)+K(k+1)v(k+1)\\ =\hat{x}(k+1|k)+K(k+1)\left[\begin{array}{cccc} H(k+1) & 0 & 0 & 0\\ 0 & H^{\left[2\right]}(k+1) & 0 & 0\\ 0 & 0 & \ddots & 0\\ 0 & 0 & 0 & H^{\left[r\right]}(k+1) \end{array}\right]\left[\begin{array}{c} \tilde{x}(k+1|k)\\ {\tilde{x}}^{\left[2\right]}(k+1|k)\\ \vdots \\ {\tilde{x}}^{\left[r\right]}(k+1|k) \end{array}\right]\\ \;+K(k+1)v(k+1)\\ =\hat{x}(k+1|k)+K(k+1)H(k+1)\tilde{x}(k+1|k)\\ \;+\sum_{i=2}^{r}K(k+1)H^{\left[i\right]}(k+1){\tilde{x}}^{\left[i\right]}(k+1|k)+K(k+1)v(k+1) \end{array}$$

The original systematic error covariance matrix is:(41)$$\begin{array}{l} P(k+1|k+1)=p_{n}\bar{P}(k+1|k+1)p_{n}{}^{T}=[I_{n\times n},0_{n\times n^{2}},\cdots ,0_{n\times n^{r}}]\bar{P}(k+1|k+1)\left[\begin{array}{c} I_{n\times n}\\ 0_{n\times n^{2}}\\ \vdots \\ 0_{n\times n^{r}} \end{array}\right]\\ =[I_{n\times n},0_{n\times n^{2}},\cdots ,0_{n\times n^{r}}][I-\bar{K}(k+1)\bar{H}(K+1)]\bar{P}(k+1|k)\left[\begin{array}{c} I_{n\times n}\\ 0_{n\times n^{2}}\\ \vdots \\ 0_{n\times n^{r}} \end{array}\right]\\ =P(k+1|k)-K(k+1)\left[\begin{array}{cccc} H(k+1) & 0 & 0 & 0\\ 0 & H^{\left[2\right]}(k+1) & 0 & 0\\ 0 & 0 & \ddots & 0\\ 0 & 0 & 0 & H^{\left[r\right]}(k+1) \end{array}\right]\left[\begin{array}{c} P(k+1|k)\\ P_{2}(k+1|k)\\ \vdots \\ P_{r}(k+1|k) \end{array}\right]\\ =P(k+1|k)-K(k+1)H(k+1)P(k+1|k)-\sum_{i=2}^{r}K(k+1)H^{\left[i\right]}(k+1)P_{i}(k+1|k) \end{array}$$

After projecting through the projection matrix, it can be seen that more information $\sum_{i=2}^{r}K(k+1)H^{\left[i\right]}(k+1){\tilde{x}}^{\left[i\right]}(k+1|k)$ is used in the original systematic state estimated value, and the item $\sum_{i=2}^{r}K(k+1)H^{\left[i\right]}(k+1)P_{i}(k+1|k)$ is subtracted from the original systematic error covariance matrix so that the error covariance matrix is reduced. Therefore, it not only retains the state estimated value of the original system and improves the estimated accuracy, but also reduces the complexity and calculation amount of filtering.

## 4. Simulation Experiment

In this section, the effectiveness of the proposed method is demonstrated through three examples.
(1)The case where the nonlinear system consists of the accumulation of several nonlinear functions [28].
(42)$$\left\{ \begin{array}{l} x_{1}(k+1)=x_{1}(k)-x_{2}(k)-\frac{1}{6}x_{1}^{3}(k)-\frac{1}{6}x_{2}^{3}(k)+\frac{1}{120}x_{1}^{5}(k)+\frac{1}{120}x_{2}^{5}(k)+w_{1}(k)\\ x_{2}(k+1)=1-\frac{1}{2}x_{1}^{2}(k)-\frac{1}{2}x_{2}^{2}(k)+\frac{1}{24}x_{1}^{4}(k)+\frac{1}{24}x_{2}^{4}(k)+w_{2}(k)\\ y(k+1)=x_{1}(k+1)+x_{2}(k+1)-\frac{1}{6}x_{1}^{3}(k+1)-\frac{1}{6}x_{2}^{3}(k+1)\\ \;-\frac{1}{2}x_{1}^{2}(k+1)x_{2}(k+1)-\frac{1}{2}x_{1}(k+1)x_{2}^{2}(k+1)+v(k+1) \end{array}\right.$$
where process noise and measurement noise satisfy the following statistical properties: w(k) and v(k) are uncorrelated zero mean noise, and $w_{1}(k)∼N(0,0.01)$, $w_{2}(k)∼N(0,0.01)$, $v_{1}(k+1)∼N(0,0.01)$, and $v_{2}(k+1)∼N(0,0.01)$. The process noise variance Q=diag(0.01,0.01), and the measurement noise variance R=diag(0.01,0.01). The initial values of the system are $x(0)={\left[1,1\right]}^{T}$, $P(0)={Ι}_{2\times 2}$. Figure 1 and Figure 2 show the real value, the estimated value of EKF, the estimated value of the second-order extended dimension, and the estimated value of the third-order extended dimension for the state $x_{1}$ and state $x_{2}$, respectively. Figure 3 and Figure 4 show the estimated error of EKF, the estimated error of the second-order extended dimension, and the estimated error of the third-order extended dimension, respectively. Table 1 shows the error comparison between EKF and the extended dimension methods.
(2)The case where a nonlinear system is multiplied by several nonlinear functions [31].
(43)$$\{\begin{array}{l} x_{1}(k+1)=0.5x_{2}(k)\sin(x_{1}(k))+w_{1}(k)\\ x_{2}(k+1)=-0.5x_{1}(k)\sin(x_{2}(k))+w_{2}(k)\\ y_{1}(k+1)=x_{2}(k+1)+v_{1}(k+1)\\ y_{2}(k+1)=x_{1}(k+1)e^{x_{1}(k+1)}+v_{2}(k+1) \end{array}$$
where process noise and measurement noise satisfy the following statistical properties:w(k) and v(k) are uncorrelated zero mean noise, and $w_{1}(k)∼N(0,0.01)$, $w_{2}(k)∼N(0,0.01)$, $v_{1}(k+1)∼N(0,0.01)$, and $v_{2}(k+1)∼N(0,0.01)$. The process noise variance Q=diag(0.01,0.01), and the measurement noise variance R=diag(0.01,0.01). The initial values of the system are $x(0)={\left[1,1\right]}^{T}$, $P(0)={Ι}_{2\times 2}$. Figure 5 and Figure 6 show the real value, the estimated value of EKF, the estimated value of the second-order extended dimension, and the estimated value of the third-order extended dimension for the state $x_{1}$ and state $x_{2}$, respectively. Figure 7 and Figure 8 show the estimated error of EKF, the estimated error of the second-order extended dimension, and the estimated error of the third-order extended dimension, respectively. Table 2 shows the error comparison between EKF and the extended dimension methods.
(3)The case where a nonlinear system is accumulated by several nonlinear functions, each of which can be multiplied by a nonlinear function.
(44)$$\{\begin{array}{l} x_{1}(k+1)=-0.85x_{1}(k)+0.5x_{2}(k)\sin(x_{1}(k))+w_{1}(k)\\ x_{2}(k+1)=-0.5x_{1}(k)\sin(x_{2}(k))+w_{2}(k)\\ y_{1}(k+1)=x_{1}(k+1)+v_{1}(k+1)\\ y_{2}(k+1)=x_{1}(k+1)e^{x_{1}(k+1)}+x_{2}(k+1)+v_{2}(k+1) \end{array}$$
where process noise and measurement noise satisfy the following statistical properties:w(k) and v(k) are uncorrelated zero mean noise, and $w_{1}(k)∼N(0,0.01)$, $w_{2}(k)∼N(0,0.01)$, $v_{1}(k+1)∼N(0,0.01)$, and $v_{2}(k+1)∼N(0,0.01)$. The process noise variance Q=diag(0.01,0.01), and the measurement noise variance R=diag(0.01,0.01). The initial values of the system are $x(0)={\left[1,1\right]}^{T}$, $P(0)={Ι}_{2\times 2}$. Figure 9 and Figure 10 show the real value, the estimated value of EKF, the estimated value of the second-order extended dimension, and the estimated value of the third-order extended dimension for the state $x_{1}$ and state $x_{2}$, respectively. Figure 11 and Figure 12 show the estimated error of EKF, the estimated error of the second-order extended dimension, and the estimated error of the third-order extended dimension, respectively. Table 3 shows the error comparison between EKF and the dimension expansion methods.

For Case 1, when the dimension of the designed filter is extended to the second order, the estimation accuracy is improved by 25.2% compared with EKF, and when the dimension is extended to the third order, it is improved by 35.8%. For Case 2, when the designed filter is extended to the second order, the estimation accuracy is improved by 89.6% compared with EKF, and when it is extended to the third order, it is improved by 94.6%. For Case 3, when the designed filter is extended to the second order, the estimation accuracy is improved by 76.5% compared with EKF, and when it is extended to the third order, it is improved by 89.5%. All three cases show that the proposed filter has better filtering effect and is more suitable for complex strongly nonlinear systems.

## 5. Conclusions and Future

A novel Kalman filter based on the Kronecker product is proposed for dynamic stochastic systems composed of the polynomial nonlinear state model and measurement model. Compared with the existing polynomial Kalman filter (PEKF) and high-order extended Kalman filter (HEKF), although PEKF can achieve higher estimated accuracy, its complexity is also high. While HEKF reduces the complexity, the estimated accuracy is also reduced. The novel Kalman filter proposed in this paper can achieve satisfactory estimated accuracy, and its problem complexity is not high. Firstly, the first-order Taylor expansion of the systematic equation is performed to obtain the Jacobian matrix and fixed deviation term. After the linearized form is obtained, the Kronecker product operation is performed on each item, and the operation of extended dimension is achieved for the state equation and the measurement equation. Finally, after projecting through the projection matrix, only the state estimated value of the original system is retained. Because more model information is used, the estimated accuracy is improved. The effectiveness of the filter is verified by simulation.

However, the filter proposed in this paper has some shortcomings. The linear term and the noise term after the approximate linearized expansion are respectively subjected to Kronecker product operation, which leads to an artificial act in describing the expanded state model error. It needs further optimization. In future research, we hope to combine the linear term and the noise term based on this paper and perform the Kronecker product operation together. A rigorous derivation will be given for the extended dimension of the modeling errors, and at the same time, the performance of the filter will be further improved.

## Figures and Tables

**Figure 1 sensors-23-02894-f001:**
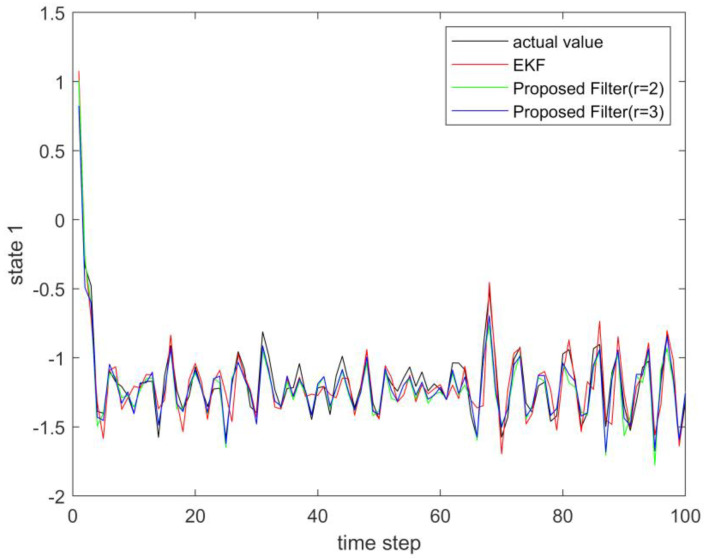
The actual value of state 1 and its estimation.

**Figure 2 sensors-23-02894-f002:**
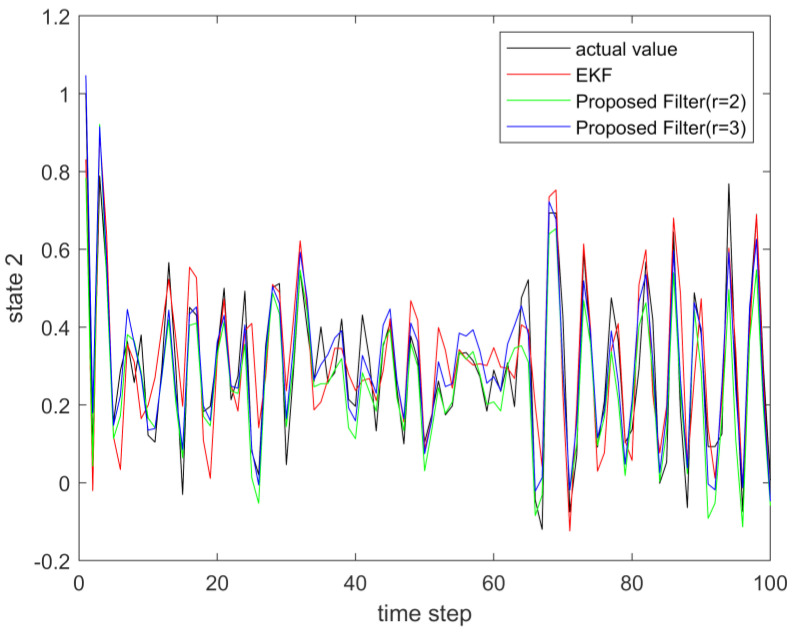
The actual value of state 2 and its estimation.

**Figure 3 sensors-23-02894-f003:**
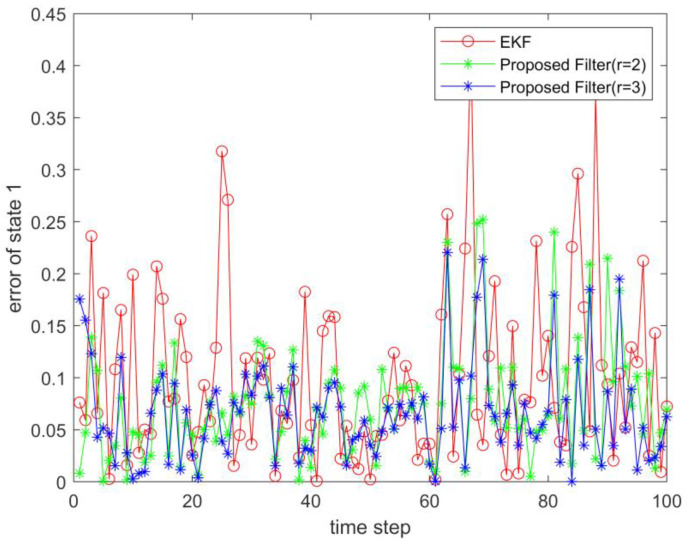
Estimation error of state 1.

**Figure 4 sensors-23-02894-f004:**
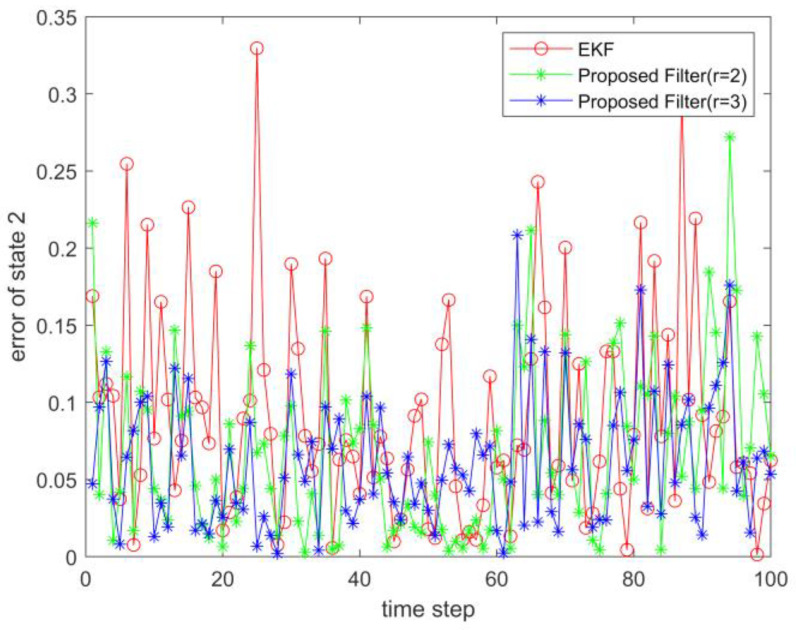
Estimation error of state 2.

**Figure 5 sensors-23-02894-f005:**
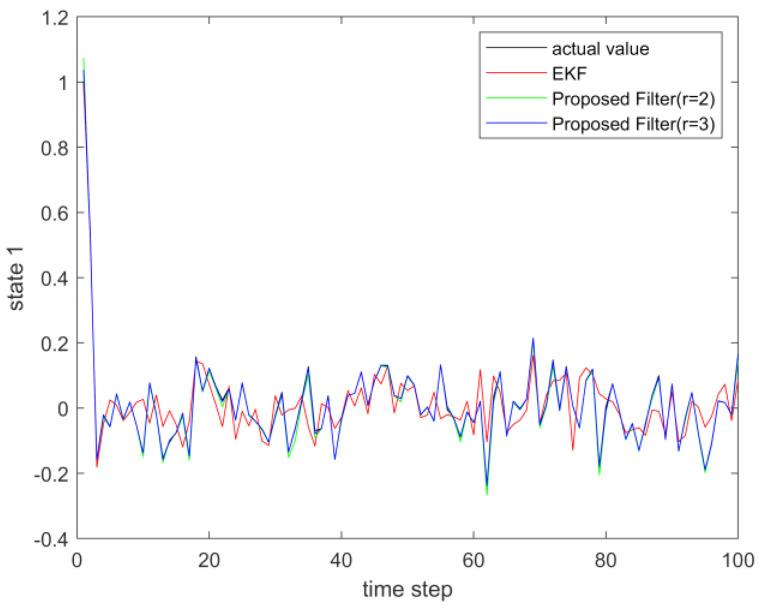
The actual value of state 1 and its estimation.

**Figure 6 sensors-23-02894-f006:**
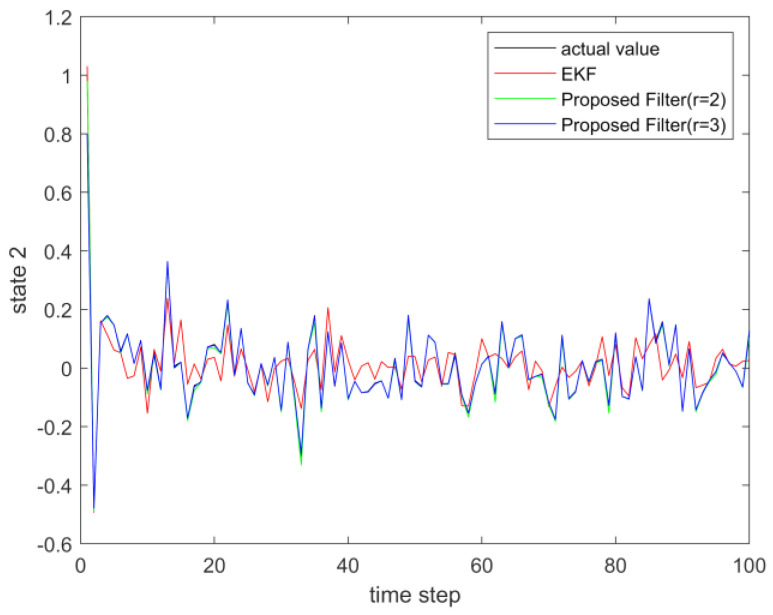
The actual value of state 2 and its estimation.

**Figure 7 sensors-23-02894-f007:**
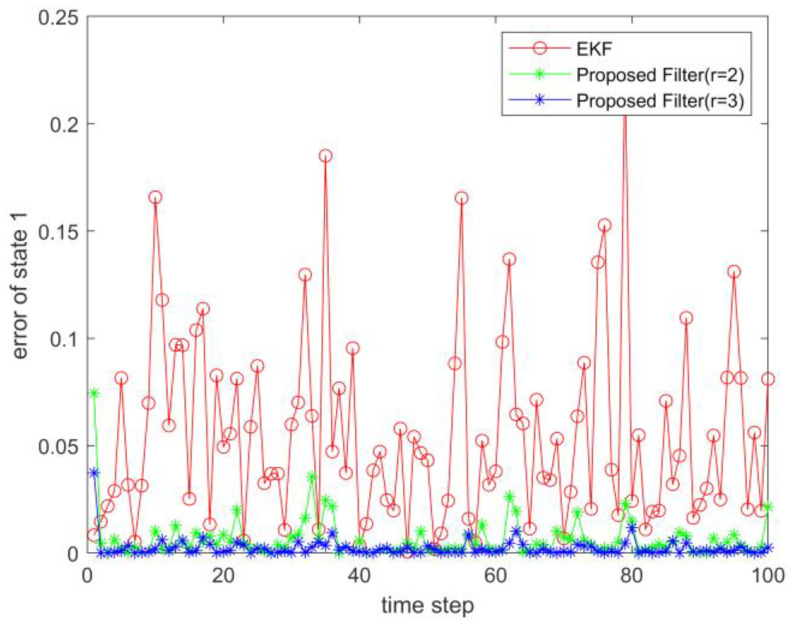
Estimation error of state 1.

**Figure 8 sensors-23-02894-f008:**
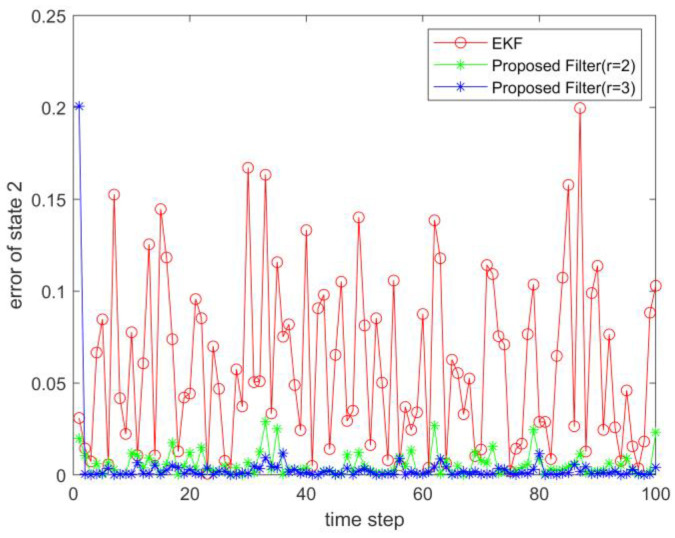
Estimation error of state 2.

**Figure 9 sensors-23-02894-f009:**
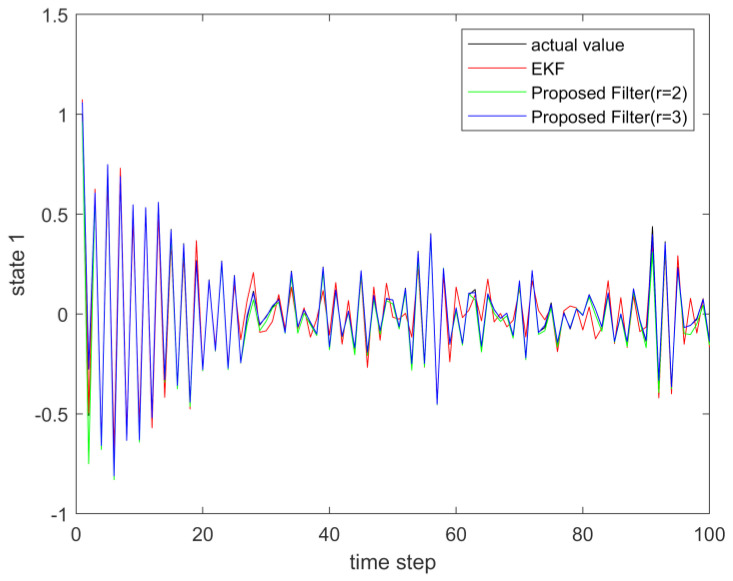
The actual value of state 1 and its estimation.

**Figure 10 sensors-23-02894-f010:**
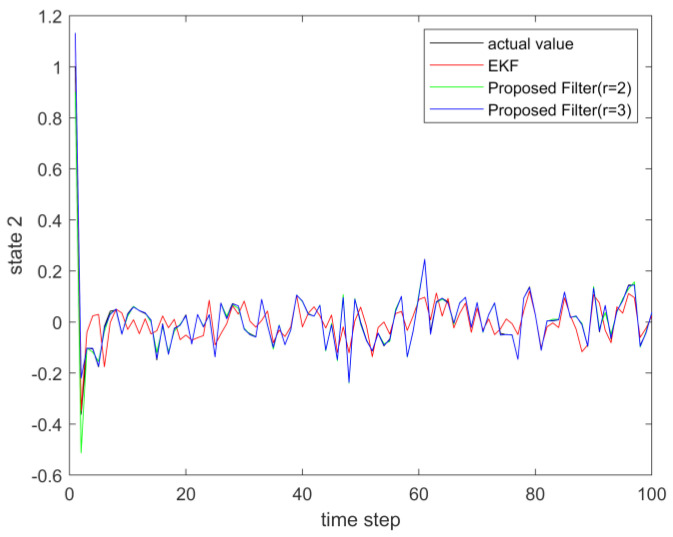
The actual value of state 2 and its estimation.

**Figure 11 sensors-23-02894-f011:**
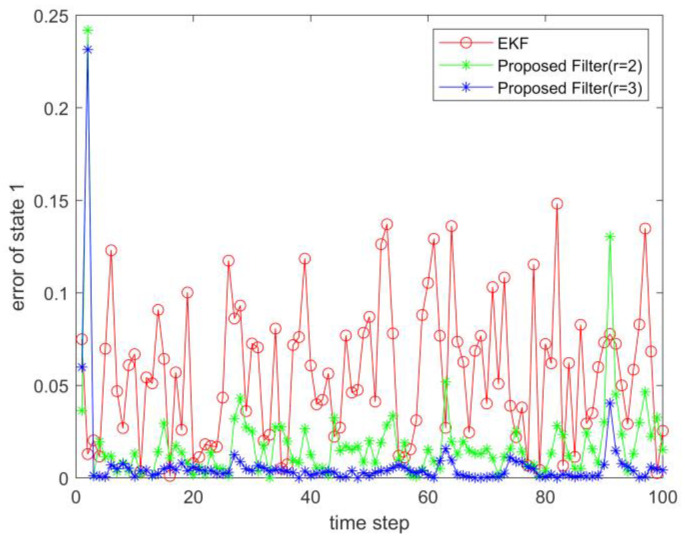
Estimation error of state 1.

**Figure 12 sensors-23-02894-f012:**
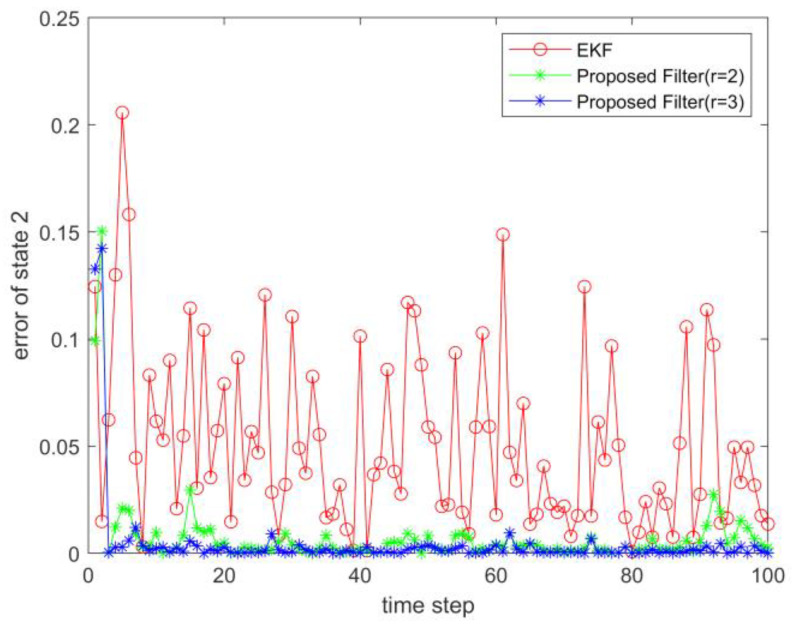
Estimation error of state 2.

**Table 1 sensors-23-02894-t001:** Error comparison between the proposed filter and EKF in case 1.

	EKF	Proposed Filter (r = 2)	Proposed Filter (r = 3)
$\mathrm{MAE}\;\mathrm{of}\;x_{1}$	0.1009	0.0746	0.0659
$\mathrm{Improved}\;\mathrm{of}\;x_{1}$(%)	/	26.1%	34.7%
$\mathrm{MAE}\;\mathrm{of}\;x_{2}$	0.0908	0.0687	0.0610
$\mathrm{Improved}\;\mathrm{of}\;x_{2}$(%)	/	24.3%	32.8%
Improved of x(%)	/	25.2%	33.8%

**Table 2 sensors-23-02894-t002:** Error comparison between the proposed filter and EKF in case 2.

	EKF	Proposed Filter (r = 2)	Proposed Filter (r = 3)
$\mathrm{MAE}\;\mathrm{of}\;x_{1}$	0.0553	0.0064	0.0023
$\mathrm{Improved}\;\mathrm{of}\;x_{1}$(%)	/	88.4%	95.9%
$\mathrm{MAE}\;\mathrm{of}\;x_{2}$	0.0595	0.0055	0.0040
$\mathrm{Improved}\;\mathrm{of}\;x_{2}$(%)	/	90.7%	93.3%
Improved of x(%)	/	89.6%	94.6%

**Table 3 sensors-23-02894-t003:** Error comparison between the proposed filter and EKF in case 3.

	EKF	Proposed Filter (r = 2)	Proposed Filter (r = 3)
$\mathrm{MAE}\;\mathrm{of}\;x_{1}$	0.0564	0.0184	0.0070
$\mathrm{Improved}\;\mathrm{of}\;x_{1}$(%)	/	67.3%	87.6%
$\mathrm{MAE}\;\mathrm{of}\;x_{2}$	0.0515	0.0073	0.0045
$\mathrm{Improved}\;\mathrm{of}\;x_{2}$(%)	/	85.7%	91.3%
Improved of x(%)	/	76.5%	89.5%

## Data Availability

Not applicable.

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
