# Peer review of "Design of a High-Order Kalman Filter for State and Measurement of A Class of Nonlinear Systems Based on Kronecker Product Augmented Dimension"

_sensors, 2023, doi:10.3390/s23062894_

Round 1

Reviewer 1 Report

The current state of the paper is essentially unreadable and extremely hard to understand. The writing needs extensive and full editing since the english is completely undeciphrable. Hence I recommend the paper be rejected to give the authors as much time as needed to re-write the whole paper so it can be essentially readable.

Reviewer 2 Report

This contribution presents original ideas in the study and advances the previous research in this area. The level of the
originality of contribution to the existing knowledge with an emphasis on the paper's innovativeness i n both theory
development and methodology used in the study is very high. This work makes a significant practical contribution and it makes
impact on the research work on the research community. The quality of arguments, the critical analysis of concepts,
theories and findings, and consistency and coherency of debate are well addressed in this paper.
The paper has a good writing style in term of accuracy, clarity, readability, organization, and formatting.

Nevertheless the following issues should be addressed:

- One of the most crucial issues is the tuning of Kalman Filter. How did you tune your Kalman Filter?

- Try to make more clear the motivation to use the Kronecker product in the proposed approch

- Please imporve the figure in terms of lables and legendas because it s not possible to read them.
The following literature can help the reader to find a more wide background on this topic.

A switching Kalman Filter for sensorless control of a hybrid hydraulic piezo actuator using MPC for camless internal combustion engines P Mercorelli 2012 IEEE International Conference on Control Applications, 980-985 28 2012
An extended Kalman filter as an observer in a control structure for health monitoring of a metal–polymer hybrid soft actuator M Schimmack et al. IEEE/ASME Transactions on Mechatronics 23 (3), 1477-1487

Alfredo Germani; Costanzo Manes; Pasquale Palumbo. Polynomial Extended Kalman Filtering[J]. IEEE Transactions on Automatic Control, 2005, 50(12).

Germani A; Manes C; Palumbo P. Polynomial Extended Kalman Filtering for discrete-time nonlinear stochastic systems[C]. IEEE Conference on Decision & Control. IEEE, 2003.

Sun Xiaohui; WEN Chenglin; WEN Tao. High-Order Extended Kalman Filter Design for a Class of Complex Dynamic Systems with Polynomial Nonlinearities[J]. Chinese Journal of Electronics, 2021, 30(03):508-515.

Reviewer 3 Report

The paper describes the design of a high-order Kalman Filter for the state and measurement of a class of nonlinear systems based on knocker product augmented dimension. Comments and suggestions are as follows:

1.      The aberrative HEKF is not assigned.

2.      Unify the equation font size with the text font size. Equations have bigger font sizes.

3.      The filter derivation is an ordinary KF approach with the augmented dimension with state matrices. Is there any theoretical proof that such an approach improves estimation –‘filtering’?

4.      Simulation introduces two imaginary dynamic models. Some additional step, such as the argument system, is missing. It would be applicable to a reader.

5.      Why is the proposed KF compared to the EKF? As you mentioned in the introduction, you have proposed the new KF, which is a tradeoff between PEKF and HEKF. Also, UKF and CKF has second-order approximation and, therefore, better-filtering property of a class of nonlinear systems. 

Reviewer 4 Report

The article deals with the application of the Kalman filter and solves the shortcomings of the existing filter. For control systems and measurement systems, the application of such types of filters is essential, so I consider the research and development of such filters to be current and important. For technical applications, the extended Kalman filter and other types are often used, which are also analyzed in this work.
The introduction of the article analyzes in detail the current state in this area and presents problems related to the application of the Kalman filter. Next is a brief theoretical introduction to nonlinear systems.
In the next part, the design idea and design method steps of the Kalman filter based on the Kronecker product are presented. The processing of this part is technically and mathematically correctly presented.

The proposed type of filter is verified by simulation experiments, which confirmed the correctness of the proposed solution. The proposed filter type is analyzed from the point of view of the informational use of the state estimation value and the measurement of the size of the covariance matrix of the state estimation errors. The effectiveness of the proposed methodology is presented on the solution of three examples. Comparisons between the errors of the extended Kalman filter and the proposed filter are given. The comparison shows better results when using the proposed filter in all three documented examples.

Comments:
The introduction of the article lacks a more detailed literary overview of similar works with relevant references to articles published in renowned journals.
All formulas, equations and mathematical symbols are written in a larger font than normal text. It looks wrong and needs to be aligned with the article template.
The conclusion of the article is insufficient. The conclusion of the article must contain a discussion and critical evaluation of the achieved results in comparison with other existing works. It is clear that the article brings new knowledge, but in the end it must be clearly defined as the novelties and main contribution of this work.
Also specify the weaknesses of the proposed filter type and the possibilities for improvement.

Reviewer 5 Report

The article deals with ensuring the accuracy of system state estimation for high-order Kalman filters. Particularly, the authors proposed to use Kronecker product augmented dimension to solve this problem for nonlinear systems. The practical significance of the presented research is in its possible wide application, primarily for real nonlinear systems when the application of Kalman filters is limited. However, the following flaws should be eliminated before the publication of the presented research:

1. Good quality of the obtained results was presented quantitatively before the conclusions. However, compared with their results only. However, a deeper discussion should be added to compare the obtained results with not only an extended Kalman filter but also with an unscented Kalman filter and other publications.

2. A significant number of articles on Kronecker product based modeling of Kalman filters for nonlinear systems were already published in 2022-2023 worldwide. In this regard, the particular scientific novelty of the presented approach should be highlighted more transparently.

3. Unfortunately, the authors did not provide any expectations on reducing the computational difficulties by filtering.

4. Moreover, the suitability of which filter to use depends on the nonlinearity of the process and the observation model. Unfortunately, this issue and the corresponding analysis were not discussed in the manuscript.

5. Despite a considerable number of up-to-date research works on Kalman filters (2022–2023), many references are outdated.

Round 2

Reviewer 5 Report

The authors have improved their manuscript properly according to all the recommendations and suggestions. Therefore, the manuscript can be considered for possible publication.